# Supernotes: Driving Consensus in Crowd-Sourced Fact-Checking

## ABSTRACT

X's Community Notes, a crowd-sourced fact-checking system, allows users to annotate potentially misleading posts. Notes rated as helpful by a diverse set of users are prominently displayed below the original post. While demonstrably effective at reducing misinformation's impact when notes are displayed, there is an opportunity for notes to appear on many more posts: for 91% of posts where at least one note is proposed, no notes ultimately achieve sufficient support from diverse users to be shown on the platform. This motivates the development of Supernotes: AI-generated notes that synthesize information from several existing community notes and are written to foster consensus among a diverse set of users. Our framework uses an LLM to generate many diverse Supernote candidates from existing proposed notes. These candidates are then evaluated by a novel scoring model, trained on millions of historical Community Notes ratings, selecting candidates that are most likely to be rated helpful by a diverse set of users. To test our framework, we ran a human subjects experiment in which we asked participants to compare the Supernotes generated by our framework to the best existing community notes for 100 sample posts. We found that participants rated the Supernotes as significantly more helpful, and when asked to choose between the two, preferred the Supernotes 75.2% of the time. Participants also rated the Supernotes more favorably than the best existing notes on quality, clarity, coverage, context, and argumentativeness. Finally, in a follow-up experiment, we asked participants to compare the Supernotes against LLM-generated summaries and found that the participants rated the Supernotes significantly more helpful, demonstrating that both the LLM-based candidate generation and the consensus-driven scoring play crucial roles in creating notes that effectively build consensus among diverse users.

ACM Reference Format:
Anonymous Author(s). 2025. Supernotes: Driving Consensus in Crowd-Sourced Fact-Checking. In *Proceedings of The Web Conference (WebConf '25)*. ACM, New York, NY, USA, 11 pages. https://doi.org/XXXXXXX.XXXXXXX

## 1 INTRODUCTION

Misinformation has become a pervasive characteristic of discourse online, leading to widespread negative consequences. Addressing its spread is a complex challenge that requires a range of mitigation strategies. Platforms have experimented with various approaches including professional fact-checking, domain filtering, and encouraging attention to accuracy [25, 30, 31]. One approach

**Figure 1: An example post, along with three community notes proposed to provide additional context, and the Supernote produced by our framework.**

that has recently attracted considerable attention is crowd-sourced fact-checking. Multiple academic studies have demonstrated that a panel of regular users could be as effective at fact-checking misinformation as professional fact-checkers [2, 4, 36].

Motivated by these findings, X has developed and launched a crowd-sourced fact-checking system called Community Notes [42]. The system allows users to create notes that can be attached to potentially misleading posts and to rate the helpfulness of proposed notes. It uses a matrix factorization algorithm to score the overall helpfulness of the notes based on the individual ratings. Notes rated helpful by many users with diverse views, as measured by their latent representations, are scored higher. Only notes that cross a certain helpfulness threshold are considered helpful and attached to the post. As of October 2, 2024, users have added more than 1.28 million notes on over 738,000 posts, and more than 78.8 million ratings. X provides continuous public access to the code and data behind the system.

Both internal pilot experiments and external analyses find that community notes have a significant impact on users' engagement with and perceptions of misinformation [6, 7, 35, 42]. Once a community note is attached to a post, the post is more likely to be deleted and less likely to be reposted or liked. Users are also less inclined to agree with the substance of misleading posts when presented with a community note [42] and tend to trust community notes more than warnings by third-party fact-checkers [10].

While the community notes have a significant impact on the posts to which notes are attached, only 12.5% of all posts for which

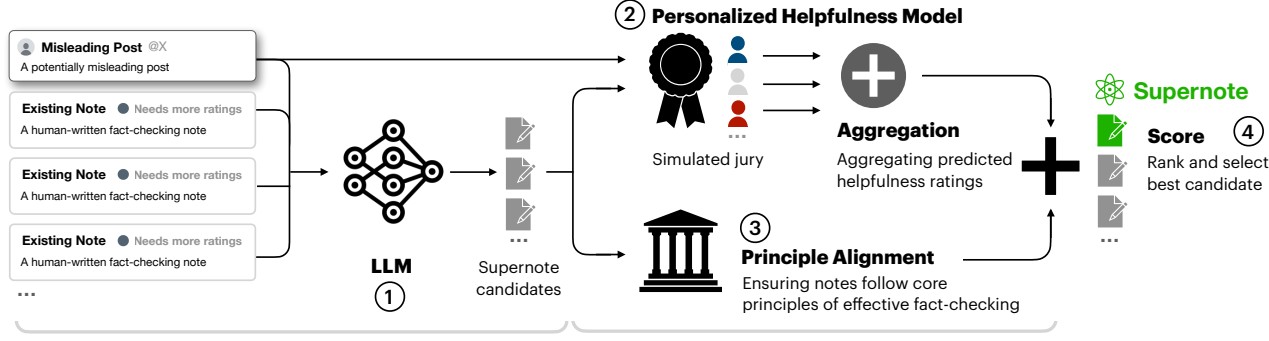

**Figure 2: Overview of our framework for generating Supernotes. (1) We prompt an LLM to generate many candidate Supernotes using the post text and the existing community notes. (2) We score the helpfulness of each candidate by simulating a jury of raters, predicting their ratings, and aggregating them using the Community Notes algorithm. (3) We filter out candidates that do not follow key principles of effective fact-checking. (4) Finally, we rank and select the candidate with the highest score.**

a note has been proposed have a note that has reached a helpful status and thus has been attached to the post. For 91% of the posts for which a note has been proposed, the note(s) have failed to gather enough helpful ratings by users with diverse views and are never attached to the post. Although some of the proposed notes may be inaccurate, many contain valuable information that offers important additional context to the post.

We postulate two reasons why otherwise valuable notes fail to reach helpful status and are never displayed on the platform. First, individual notes may present biased or one-sided perspectives, failing to achieve the neutral tone and information necessary for broad acceptance. This can lead to polarized ratings, with users aligned with the note's perspective rating it as helpful, while others find it unpersuasive or even misleading. For example, in Figure 1, the second note might be perceived as argumentative or partisan, hindering its adoption by a wider audience. Second, essential context is often fragmented across multiple notes, each offering a piece of the puzzle but failing to provide a holistic understanding in isolation. Figure 1 illustrates this, with separate notes addressing President Biden's authority on debt cancellation and the administration's proposal on medical debt reporting. These fragmented insights, while individually valuable, may not achieve sufficient visibility on their own.

**The present work.** We propose a framework for generating *Supernotes*: notes that synthesize information from several existing community notes and are written to foster consensus among a diverse set of users. The key idea behind the framework is to use LLMs to generate many candidate Supernotes and use a helpfulness model trained on millions of publicly available Community Notes ratings to select the candidates that are most likely to be rated helpful by a diverse set of users (Figure 2).

The framework consists of two components: candidate generation and candidate scoring (Section 2). The candidate generation component leverages an LLM to generate many diverse candidate Supernotes. The candidate scoring component ranks the proposed candidate by how likely it is to be rated as helpful by a diverse set of users and whether it follows core principles of effective fact-checking. To estimate whether a proposed Supernote would be rated

helpful broadly, we simulate a jury of randomly sampled raters, predict their individual ratings, and aggregate them in a single helpfulness score. We leverage the millions of publicly-available ratings on existing community notes to train a model that given the post, note, and rater information predict the rater's helpfulness rating. To aggregate the predicted ratings, we use the Community Notes matrix factorization algorithm [42]. The candidate scoring component also ensures that the Supernotes follow core principles for effective fact-checking (e.g., are unbiased and non-argumentative) and do not introduce any new links.

To test our framework, we evaluate the performance of the individual components on historical data (Section 3) and conduct human subject experiments (Section 4). At the individual rating level, we show that our personalized helpfulness model accurately predicts ratings on community notes (AUC = 0.85) that were held out during training. At the jury level, we show that aggregated predictions of a jury of raters accurately predicts which note among all notes on a held out post is most helpful according to the empirical ratings (P@1 = 0.7). To evaluate the helpfulness of the Supernotes produced by our framework, we sample 100 posts and recruit participants to rate the helpfulness of the Supernotes and the best existing note on the post, without revealing which one is which. We find that participants rate the Supernotes as significantly more helpful than the best existing note and, when asked to choose among the two, select the Supernote 75.2% of the time. When asked to rate the note's quality, clarity, coverage, context, and argumentativeness, participants rated the Supernotes significantly more favorably than the best existing notes across all five dimensions. Finally, to evaluate the impact of the Candidate Scoring component, we ran a follow-up experiment in which we asked participants to compare the Supernotes generated by the full pipeline with an LLM summary generated by our Candidate Generation component. We find that the participants rated the Supernotes as more helpful than the LLM summaries 61.5% of the time, demonstrating the impact of the Candidate scoring component.

We release the code needed to implement our framework and replicate our analyses at: https://github.com/supernotes-research/anonymous_repo/

## 2 SUPERNOTE GENERATION: FRAMEWORK

In this section, we present the proposed framework for generating Supernotes in which we leverage a generative language model guided by a synthetic jury that is simulated using a personalized helpfulness model (PHM). At a high level, the framework consists of two components: (1) *candidate generation*, which generates many Supernote candidates, and (2) *candidate scoring*, which evaluates every candidate and chooses the best one (Figure 2). In this section, we describe the key ideas behind the framework, and in Section 3 we detail our implementation and offline evaluation of the framework.

### 2.1 Candidate Generation

In the first component of the framework, we prompt an LLM to generate many Supernote candidates given the post text and random subsets of existing community notes. The goal of this component is to generate many diverse candidates that will be later scored by the scoring model. We feed the information to an LLM via a prompt that describes (1) the goal of the system, (2) the characteristics of effective Supernotes, (3) provides the post and existing notes, and (4) instructs the model to synthesize the notes into a single helpful note that adds context to the original post. The definition of effective notes closely follows the Community Notes note-writing guidelines that outline the attributes of helpful notes, such as quality of sources, relevance, ease of understanding and provision of useful context. In addition to the content of the notes, we provide auxiliary information for each note to the model, including the helpfulness ratings (helpful, somewhat helpful, or not helpful) and the aggregated tags that raters used to describe them. Finally, we set the hyperparameters of the LLM generation and sampling (e.g., temperature and top $k$) to increase diversity in the generated candidates. The goal is to sample a diverse set of possible candidates and select the most appropriate one based on the candidate scoring component.

### 2.2 Candidate Scoring

The second component of the framework scores the generated Supernote candidates. It consists of two sub-components: (1) *helpfulness scoring by a jury*, in which we simulate the helpfulness ratings by a synthetic jury of raters, and (2) *principle alignment*, in which we ensure that the candidates follow key guidelines for effective fact-checking.

**Simulated Jury: Rating Prediction.** To simulate a jury of raters, we sample individual raters, predict how they would rate a candidate Supernote, and aggregate these ratings into a single helpfulness score. We use a personalized helpfulness model (PHM), which predicts how each rater would respond based on the note content and the rater's profile. The PHM, trained on millions of Community Notes ratings, outputs probabilities for three categories: "helpful," "somewhat helpful," and "not helpful." Instead of choosing the most likely category (greedy sampling), we apply probabilistic sampling, selecting a rating based on the predicted probabilities. Greedy sampling introduces bias by over-selecting the "helpful" category, inflating the aggregated helpfulness score and reducing correlation with actual Community Notes ratings. Probabilistic sampling, in contrast, maintains the relative uncertainty of the predictions, yielding a more unbiased and accurate estimate of the

true distribution of ratings (Appendix A.1). This is possible because our model's probabilities are well-calibrated (Appendix A.2).

**Simulated Jury: Jury Sampling and Rating Aggregation.** For each Supernote candidate, we sample a random set of raters to be included in the jury. We sample uniformly at random from all raters in the Community Notes data. Other sampling strategies could also be appropriate, e.g., sampling raters that are more likely to rate the given note or sampling a more politically polarized jury to evaluate notes on exceptionally sensitive posts on political issues. Next, we predict the rating of each member of the jury using the model described above and aggregate their ratings.

There are many possible aggregation functions, ranging from simple majority voting to more sophisticated social welfare functions that map individual utilities to collective welfare [3, 20, 27]. However, given our goal of improving the Community Notes platform, we follow the Community Notes aggregation function [42]. The Community Notes algorithm factorizes the note-rating matrix and uses the calculated note intercepts to measure the helpfulness of notes [42]. Due to the mechanics of the matrix factorization, notes rated helpful by raters who typically disagree have higher note intercepts. To obtain the note intercepts (i.e., the note scores) for the candidate Supernotes, we project the predicted ratings by jury members onto the latent space inferred by the matrix factorization using least squares. We provide more details about this aggregation step in Appendix B.1.

**Principle Alignment.** In addition to ensuring that the Supernotes are likely to be rated as helpful by a diverse set of users, we also ensure that the Supernotes align with first-order principles of effective fact-checking. We enforce two key principles: (1) the Supernotes should be written in a neutral and unbiased language, and (2) the Supernotes should not express speculations and opinions. We test whether candidate Supernotes follow these principles by prompting an LLM with the definition of the principle and the content of the Supernote (Appendix B.3). We also test for two other, rule-based, criteria: (a) we ensure that the Supernotes do not introduce any new links that are not already present in the existing notes and that the links included are valid; and (b) we ensure that the Supernotes (excluding links) are not longer than 280 characters, the maximum length allowed by Community Notes. If a candidate Supernote does not satisfy any of the above criteria, it is excluded from the pool of candidate Supernotes rated by the simulated jury.

## 3 SUPERNOTE GENERATION: IMPLEMENTATION

In the previous section, we described the key ideas behind the Supernote generation framework; next, we present our particular implementation choices and offline evaluation of the key components of the framework.

### 3.1 Candidate generation

To generate Supernote candidates given the post and the existing community notes, we used OpenAI's GPT-4o-mini model. For each post, we generate 100 Supernote candidates. To promote diversity in the generated candidates, we prompt the LLM with different permutations of subsets of eligible existing notes and set the *temperature* = 0.95, *top_p* = 0.8. We only consider notes that

(1) argue that the post is misleading, and (2) have a "Needs More Ratings" status, i.e., they do not have enough "helpful" ratings to be deemed "Currently Rated Helpful" and do not have enough "not helpful" ratings to be considered "Currently Rated Not Helpful." More details on the specific note-selection criteria in our evaluation setup is described in Section 4.1. To decide between different prompt templates, we (1) sampled a random set of posts, (2) applied our simulated jury and principle alignment framework to choose the best candidate, and (3) examined the distribution of Supernote helpfulness scores and the frequency of principle misalignment. We then chose the prompt template with the highest average Supernote helpfulness score and lowest principle misalignment frequency. We iterated on the prompt while observing the outputs and found that the following aspects of the prompt are most important: (1) precise description of the Community Notes system, (2) explicit instructions that the Supernotes should be compelling to a diverse set of raters, (3) definition of the effective fact-checking principles, and (4) inclusion of the tags attached by raters describing the existing notes (Appendix B.3). We note that the choice of the prompt may depend on the particular language model used. For instance, we found that GPT-4o-mini followed the prompt instructions more closely than GPT-3.5, as previously observed [1, 34].

## 3.2 Candidate Scoring

**Simulated Jury: Rating Prediction.** Next, we train models that, given the post, note, and rater information, predict how a rater would rate the note. To represent the note and the post, we use OpenAI's *text-embedding-3-small* model to obtain a 512-dimensional embedding of each text string, respectively. To represent the rater, we use a 2-dimensional embedding consisting of the rater factor and intercept computed by running the Community Notes production algorithm on historical rating data. The rater intercept measures the propensity of the rater to rate notes as helpful, and the rater factor represents the rater's viewpoint [42]. The models, which take in a 1026-dimensional vector as input, are trained as classifiers with three outputs corresponding to the possible helpfulness ratings on Community Notes: "helpful", "somewhat helpful", and "not helpful".

*Ratings Data.* To train the models, we use 2.1 million publicly available Community Notes ratings. We exclude ratings collected during the pilot stage of the system (before 2023) and focus on notes in English (as detected by *langdetect*). We sort the data chronologically and split it into train (80%), validation (10%), and test (10%) sets. We remove notes with ratings that span more than one interval from later intervals to ensure that there are no spillovers of training or validation data on the test set. To compute the rater embeddings fed into the prediction models, we use the ratings available up to the end of the relevant time period to prevent any indirect information leakage via rater embeddings from the future.

*Model Evaluation.* We test a variety of models: logistic regression, ridge regression, random forest, and a neural network. For reference, we also include a dummy classifier that always predicts the most frequent class. We find that the neural network significantly outperforms the other models across various metrics (precision, recall, F1, and AUC) and achieves an AUC of 0.85 (Figure 3). The neural network had ten progressively smaller layers and was trained using a cross-entropy loss with the Adam optimizer and a learning rate

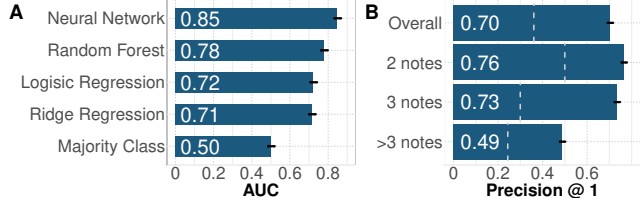

**Figure 3: (A) Out-of-sample performance of the classification models predicting whether a rater would rate a note as "helpful", "somewhat helpful", or "not helpful" given the post, note, and rater embeddings. (B) Precision of our simulated jury—which aggregates the individual predicted helpfulness ratings—in predicting the most helpful note according to the empirical Community Notes ratings on off-sample historical posts. The dashed vertical lines represent the theoretical baseline of random guessing. The error bars represent 95% CIs.**

of $10^{-5}$. We ran training for 20 epochs with mini-batches of size 32. We selected the hyperparameters by tuning one parameter at a time until we did not observe any improvements on the validation set.

As we describe in Section 2.2, to reflect the relative difference between the predicted probabilities in the aggregate scores for each note, it is critical that the model makes predictions by sampling proportionally to the predicted probabilities for each class instead of predicting the most likely class. Empirically, we find that this modification significantly reduces the mean absolute error between the predicted note helpfulness scores based on our simulated juries and the ground-truth scores computed using the Community Notes algorithm with the observed ratings (Appendix A.1).

**Simulated Jury: Jury Sampling and Rating Aggregation.** Next, we determine the helpfulness of the note by aggregating the rating predictions of individual jurors using the Community Notes matrix factorization algorithm (Appendix B.1). While we demonstrated that the individual rater predictions tend to be accurate, it might be that their errors accumulate when aggregated and lead to less accurate predictions of note helpfulness. In a production setting our framework would need to identify whether the Supernote is more helpful than the existing notes, and if so, display it on the platform. To match this task, we test whether the aggregated predictions can accurately identify the most helpful note among the proposed notes on the post according to the observed Community Notes ratings. We predict the scores for a random sample of 2,000 notes from the test set by aggregating the predicted ratings of individual jurors. Then, we measure the precision with which the highest scoring note (as determined by aggregating predicted helpfulness ratings from our simulated jury) matches the note with the highest Community Notes score (as computed by the Community Notes scoring algorithm using all observed ratings). We find that our predictions have a Precision@1 of 0.7 across all posts, 1.94 times higher than guessing at random (Figure 3B). As expected, the precision of our predictions is largest when there are only two proposed notes on the post (P@1 = 0.76) and lower when there are three (P@1 = 0.73) or more notes (P@1 = 0.49).

**Principle Alignment.** Among all the principles tested for alignment, link invalidity had the highest rejection rate at 8%. These

invalid links typically include a truncated portion of a valid link or, on rare occasions, are entirely hallucinated by the LLM. We note that using GPT-4o-mini for candidate generation significantly reduces the prevalence of invalid links compared to GPT-3.5 Turbo, which yields a rejection rate of 22%. The rejection rates for argumentativeness and lack of clarity were small, less than 2%, likely because we include these principles in the prompt.

## 4 HUMAN EVALUATION

Following the promising results of the offline evaluation of our framework, we ran a human subjects experiment to evaluate the helpfulness of the generated Supernotes against the best existing community notes. In this section, we describe the results of our main experiment and follow-up ablation study, evaluating the importance of our personalised helpfulness model. The experiments were approved by our institution's IRB.

### 4.1 Experimental Setup

The goal of this experiment is to compare the helpfulness of Supernotes with the helpfulness of the best existing notes on Community Notes and test whether Supernotes adhere more strongly to the principles of effective fact-checking defined by the platform.

**Survey Structure.** In the main part of the experiment, participants were asked to evaluate the Supernotes and the best existing notes on 12 posts from X. We presented one post at a time in a random sequence, with the two notes also displayed in random order. The participants were unaware of which note was the Supernote and which was the best existing note. For each note, the participants were asked to:

(1) Rate the helpfulness of the notes on a three-point scale: "not helpful", "somewhat helpful", or "helpful", following the labels available on Community Notes platform;
(2) Rate their agreement with five statements related to key principles of effective fact-checking (e.g., whether the note is written in a clear language) on a five-point Likert scale; the statements were inspired by the Community Notes tags (Appendix B.2) and aim to provide further context to the participants' ratings;
(3) Choose which one of the two notes is more helpful.

To guide participants and maintain consistency, we began the experiment with a brief overview of the key characteristics of effective notes, inspired by the Community Notes writing tips. To discourage participants from copying text and using generative AI tools when responding, we showed the screenshots of the posts. The notes were displayed in plain text and the participants could follow the links included.

In the final section of the survey, the participants were shown ten additional posts along with a single existing community note for each post. These notes were selected to have large note factors in the Community Notes matrix factorization, indicating that they have received significantly different helpfulness ratings from users who usually have opposing views. We also restricted this sample to English posts related to US politics. The participants were only asked to rate the notes' helpfulness ("not helpful", "somewhat helpful", and "helpful"). We utilized these ratings to anchor participants'

factors within the Community Notes matrix factorization and calculate the Community Notes helpfulness scores for both Supernotes and the best existing community notes, based on their responses.

**Post and Note Selection.** Each participant was shown a random sample of 12 posts from a pool of 100 posts selected for the experiment. We designed the post selection process to mimic the setup in which our framework would be deployed. For each post, we set a time cutoff of one hour after the third note had been posted. We then apply the Community Notes algorithm to calculate the helpfulness score for each note based on the empirical ratings up to the cutoff and select the highest-scoring note (i.e., best existing note) for comparison with the Supernote. We also use the posts' notes and their ratings up to the cutoff to attempt to generate a Supernote. We consider only notes that argue that the post is misleading and the Community Notes algorithm classifies as "Needs More Ratings", i.e., have not received enough positive/negative ratings to be scored as helpful/unhelpful. If one of the notes reaches a helpful status, we do not consider that post as the Community Notes users have already converged on a helpful note and a Supernote will not add any value. To select the posts, we also require that the Supernote is scored higher than the best existing notes and above 0.4 (the threshold in the Community Notes algorithm) according to our scoring model. We considered only posts created up to a month before the experiment. To avoid any language, length, and media effects, we further restricted the pool to English posts shorter than 280 characters that did not contain videos, but may contain images (59%) or external links (10%).

**Participants.** We used CloudResearch Connect to recruit participants. The sampling frame was restricted to English-speaking adults living in the US. We used quotas to ensure an equal split between self-reported left and right-leaning participants. The participants were compensated at the rate of $15/hour. All user data was anonymized and stored securely, following IRB-approved guidelines. In total, we collected 1,008 ratings contributed by 42 participants. The median time for completing the survey was 25.5 minutes.

### 4.2 Results

Next, we analyze the participants' responses to investigate whether the proposed Supernotes are indeed more helpful than the best existing notes on the platform. We consider four aspects: (1) participants' helpfulness ratings, (2) the rate at which participants found the Supernotes more helpful when required to choose between them and the best existing note (win rates), (3) the helpfulness scores calculated using the Community Notes algorithm, and (4) participants' evaluations of how well the notes adhere to core principles of effective fact-checking.

**Helpfulness Ratings.** We find that when asked to rate the notes as "not helpful", "somewhat helpful", and "helpful" participants rated the Supernotes as significantly more helpful than the best existing notes (Figure 4A; Wilcoxon signed-rank test, $p < 0.001$). We also compared the helpfulness of the Supernotes with the best existing notes when mapping the helpfulness ratings to the numerical values used in the Community Notes algorithm ("not helpful": 0, "somewhat helpful": 0.5, and "helpful": 1) [42]. We find that, on average, the Supernotes were rated significantly more helpful

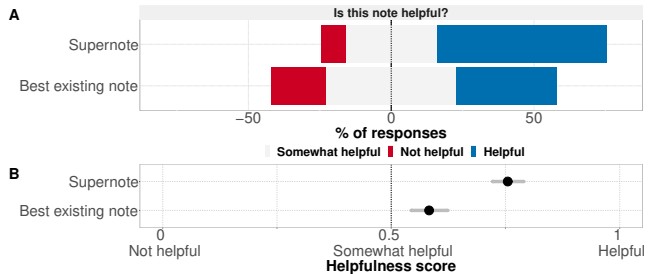

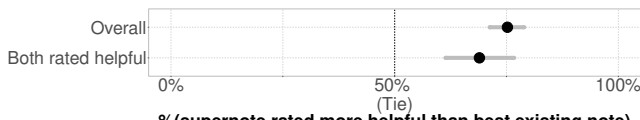

**Figure 5: Helpfulness win rates.** The rate at which participants selected the Supernote as more helpful than the best existing community note. Overall, the Supernotes were selected as more helpful 75.2% of the time and 68.9% of the time when both notes were also rated as "helpful". The error bars represent 95% CIs.

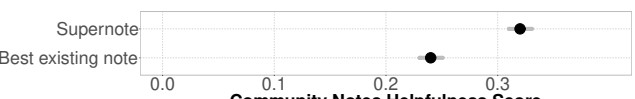

**Figure 6: Community Notes Helpfulness Scores.** Comparison between the helpfulness scores for the Supernotes and the best existing notes computed using the Community Notes algorithm. The Supernotes are scored significantly higher than the best existing notes. The error bars represent 95% CIs.

**Figure 4: Helpfulness ratings.** Comparison between the participants' helpfulness ratings of Supernotes and best existing notes on three-point scale (A) and mapping of the helpfulness ratings to numerical values used in the Community Notes algorithm (B). The Supernotes are rated as significantly more helpful than the best existing notes in both analyses. The error bars represent 95% CIs.

than the best existing notes (Figure 4B; $\mu_{Supernote}$ = 0.76, 95% CI: [0.722, 0.788]; $\mu_{best\text{-}existing}$ = 0.58, 95% CI: [0.544, 0.622]; paired t-test, $p < 0.001$). Finally, we run a linear mixed-effects regression to account for the fact that each participant rated notes on multiple posts and each post's notes were rated by multiple participants. We regress the numeric helpfulness score on a binary indicator for whether the note was a Supernote and participant and post random intercepts. We find that the coefficient of the Supernotes indicator variable is $\beta_{is\text{-}Supernote}$ = 0.17 ($p < 0.001$), i.e., the helpfulness ratings of the Supernotes are 0.17 higher than that of the best existing notes ($\beta_0$ = 0.58), providing further evidence that the Supernotes are rated as significantly more helpful.

**Helpfulness Win Rates.** While the Supernotes are rated as more helpful overall, the three-point helpfulness scale ("not helpful", "somewhat helpful", "helpful") may not have been granular enough for the participants to indicate which one of the two notes is more helpful. To examine this possibility, we also asked participants to choose which one of the two notes they found more helpful (blinded to which one was the Supernote and which one was the best existing note). We find that participants selected the Supernote as more helpful than the best existing note in 75.2% of the cases (95% CI: [0.71, 0.79]; Figure 5). Focusing only on instances where both notes were rated as "helpful," the Supernote was chosen as the more helpful 68.9% of the time (95% CI: [0.61, 0.76]). We find similar results when we fit a linear mixed-effects regression modeling the nested structure in the data. We regress an indicator variable for whether the Supernote is selected as more helpful than the best existing note on a global intercept with participant and post random intercepts. The coefficient of the global intercept is $\beta_0$ = 0.75 ($p < 0.001$), confirming that the Supernotes are preferred more often than the best existing notes.

**Community Notes Helpfulness Scores.** So far, we have demonstrated that the Supernotes were rated and chosen as more helpful than the best existing notes. However, the Community Notes algorithm scores notes based on how helpful they are rated by users who usually disagree. Next, we compare the helpfulness scores of the Supernotes and the best existing notes as calculated using the Community Notes algorithm. We compute the Community Notes

scores as follows: *(i)* we run the Community Notes algorithm on all public ratings and save the model parameters; *(ii)* we concatenate the ratings collected in our experiment on existing notes to the public ratings and re-run the Community Notes algorithm, keeping the previously computed model parameters fixed, and learning the factors and intercepts of the participants in the experiment; *(iii)* using the ratings collected in our experiment on new (super)notes, and rater parameters fixed, we learn the note parameters of the new notes. The intercepts of the new notes correspond to their helpfulness scores. We point out that the experimental data also includes the participants' ratings on the ten polarizing notes, rated by all participants and selected to be the most informative of the participants' factors. We find that the Supernotes have significantly higher helpfulness scores than the best existing notes (Figure 6; $\mu_{Supernote}$ = 0.32, 95% CI: [0.31, 0.33]; $\mu_{best\text{-}existing}$ = 0.24, 95% CI: [0.23, 0.25]; paired t-test, $p < 0.001$).

**Note Characteristics.** Finally, we investigate why the participants find the Supernotes more helpful than the best existing notes. In addition to asking participants to rate the helpfulness of the notes, we also asked them to rate their agreement with five statements about the notes related to their source quality, clarity, coverage, context, and argumentativeness (Figure 7). The statements were inspired by the Community Notes tags that users can specify when they rate notes on the platform (Appendix B.2). We find that the participants rated the Supernotes more positively than the best existing notes across all five aspects (Wilcoxon signed-rank tests, all $p < 0.001$). We observe the most significant differences in the participants' ratings on whether the notes provide important context and include high-quality sources. These findings indicate that the Supernotes effectively synthesize information from multiple notes, providing more holistic context, including more sources, and presenting the information in clear, unbiased, and non-argumentative language.

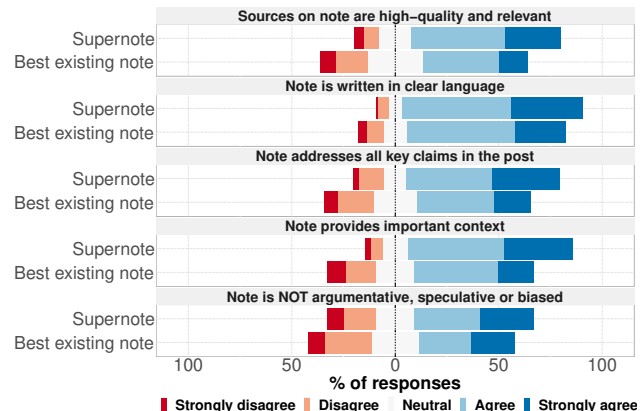

Figure 7: Note Characteristics. Participants' evaluations of how well the notes adhere to five key principles of effective fact-checking. Supernotes were rated significantly higher than the best existing notes across all five dimensions.

**Summary of the Results.** We find that the participants rated the Supernotes as more helpful than the best existing notes (Figure 4) and consistently chose the Supernotes as the more helpful when asked to select between the two (Figure 5). The Supernotes were also scored higher than the best existing notes by the Community Notes algorithm, suggesting that they are also rated helpful by participants who typically disagree (Figure 6). Lastly, participants rated the Supernotes more favorably across multiple aspects, including source quality, clarity, coverage, context, and argumentativeness (Figure 7).

### 4.3 Ablation Study: Impact of the Candidate Scoring Component

The results of the experiment presented above demonstrate that the Supernotes produced by our framework are significantly more helpful than the best existing community notes. In this section, we show that the Candidate Scoring component, which simulates a jury of raters to score and rank the candidate Supernotes, is crucial to the performance of our framework.

**Setup.** To evaluate the impact of the Candidate Scoring component, we conducted a follow-up experiment in which we asked participants to choose whether the Supernotes are more helpful than a randomly selected summary produced by the Candidate Generation component. As a reminder, the Candidate Generation component generates many candidate summaries using subsets of permutations of existing eligible notes. We randomly selected 40 posts and recruited 20 participants, evenly split between left-leaning and right-leaning individuals, as in the previous experiment. Each participant rated 20 posts and each post received 10 ratings. The Supernote and random summary were shown in random order, and participants were not informed which one was which.

**Results.** We find that the participants rated the Supernotes as more helpful than the randomly selected summaries 61.5% of the time (95% CIs: [0.57, 0.66], Figure 8). We observe similar results when we fit a mixed-effects model regressing an indicator variable for whether the Supernote was selected as more helpful on global

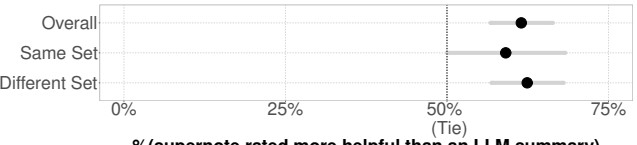

Figure 8: Effectiveness of the Candidate Scoring component. Participants found the Supernotes to be more helpful than randomly selected candidate summaries in 61.5% of cases. The Supernotes were rated more helpful both when the Supernote and the random summary were derived from the same or a different set of existing notes. This demonstrates that the Candidate Scoring component performs well in both selecting the most relevant existing notes and synthesizing their information in a helpful manner.

intercept with participant and post random intercepts ($\beta_0 = 0.63$, $p < 0.001$). These results demonstrate that the Candidate Scoring component has a significant impact on the quality of the Supernotes produced by our framework.

To further investigate the impact of the Candidate Scoring component, we break down the posts by whether the Supernote and the random summary consider the same or a different set of existing notes (Figure 8). We find that participants chose the Supernotes as more helpful 59.1% (95% CIs: [0.5, 0.68]) of the time when both the random summary and the Supernotes were based on the same set of notes. This suggests that the Candidate Scoring component effectively identifies Supernotes that convey the same information in a more helpful manner. Similarly, the participants find the Supernotes more helpful 62.4% (95% CIs: [0.57, 0.68]) of the time when the Supernotes and the random summary are based on a different set of notes, providing evidence that the Candidate Scoring component is effective in both selecting the most appropriate existing notes and synthesizing their information in a helpful way.

## 5 FURTHER RELATED WORK

We build upon a rich body of previous research focusing on scaling fact-checking and aggregating diverse perspectives using LLMs. In this section, we highlight the studies most closely related to our work.

### 5.1 Scaling Up Fact-Checking

Traditional fact-checking by human experts and organizations [16, 37] are difficult to scale up [31] and suffer from declining levels of trust from polarized social media users [13, 39]. This has prompted a lot of research on crowd-sourced [21] and automated [34] fact-checking as more scalable alternatives. Crowd-sourced fact-checking outcomes are often found to align with expert judgements [2, 36], be resilient against motivated reasoning [11], and reach more polarized online communities [26]. Although most deployments of such systems still incorporate some expert oversight [14, 29], Community Notes is a notable example of a fully crowd-sourced fact-checking system successfully implemented on a large social network [6, 9, 10, 32].

Multiple recent studies have examined how LLMs can support or even replace critical steps in the fact-checking process [18, 34], including identifying misleading claims [33], verifying information [23, 41], and generating explanations [17, 22, 43, 44]. Other research has proposed fully automating the fact-checking process by leveraging generative models' ability to handle multimodal data and query external sources [45]. Most of these systems aim to correct factually inaccurate social media posts. Our approach differs in two key ways. First, rather than querying external data sources, we ingest the proposed notes and early ratings from Community Notes contributors. While slower, this approach is less prone than AI-only approaches to unintentionally introduce errors that may quickly erode the users' trust in the system [8]. Second, we focus on generating notes that are predicted to drive consensus among users. This is important not only for ensuring that Supernotes are rated as helpful by a diverse set of users but also for maximizing their effectiveness once they are displayed on the platform.

## 5.2 Aggregating Diverse Perspectives with LLMs

While most work on LLMs has focused on improving one-on-one interactions, there is growing interest in their ability to enhance collective intelligence and democratic processes [24, 40]. LLMs can support various aspects of group decision-making, such as increasing accessibility through translation, facilitating brainstorming, mediating deliberations, and aggregating diverse perspectives [5].

Our work builds on efforts that focus specifically on aggregating perspectives [3, 12, 19, 38]. Two key papers provide the foundation for this work. First, Bakker et al. [3] used LLMs to map opinions into consensus statements by combining a generative model with a personalized preference model in a single framework. While their model estimates personalized agreement with political statements based on written individual opinions, our personalised helpfulness model conditions on the raters' factors and intercepts to predict personalized helpfulness for fact-checking notes. Additionally, we differ in our aggregation method: Bakker et al. use a simpler social welfare framework, whereas we adopt the Community Notes aggregation method to compute helpfulness scores.

Second, we draw inspiration from Fish et al. [12], who also combine LLMs with social choice but generate multiple summary statements rather than a single consensus statement. Unlike Fish et al., who rely entirely on prompted off-the-shelf models for both generative and reward tasks, we use a similar approach for the generative model but leverage millions of historical Community Notes ratings to train a custom personalized helpfulness model that incorporates the raters' factors and intercepts for more accurate predictions.

## 6 DISCUSSION AND CONCLUSION

In this study, we proposed a framework for creating Supernotes, constrained summaries of community notes designed to drive consensus among diverse users and adhere to principles of effective fact-checking. In a human subjects experiment, we demonstrated that the Supernotes produced by our framework were rated as more helpful than the best existing community notes. Aggregating the participants' ratings of each post, we also found that the Supernotes were scored higher by the Community Notes algorithm. Qualitatively, we found that the Supernotes are perceived to have

higher quality sources, be more comprehensive in coverage of key points, and be less argumentative than the best existing notes. Finally, in a follow-up experiment, we asked participants to compare the Supernotes against LLM-generated summaries. We found that the participants rated the Supernotes as significantly more helpful, demonstrating that our candidate scoring model is effective in selecting notes that drive consensus among diverse users.

**Limitations.** Our work is not without limitations. First, we ran our human subject experiments using surveys outside of the platform. Although we made considerable efforts to replicate the Community Notes user experience—using identical prompts and options to gather helpfulness ratings—it is possible that the participants might have behaved differently if they had performed the same task directly on the platform. Second, while on Community Notes users can choose which notes to rate, we required our participants to rate specific notes and compensated them for their time. To partially mitigate this limitation, we made a deliberate effort to recruit a diverse group of social media users for the study. Third, our framework relies on existing notes and ratings and, as a result, Supernotes will inherently be generated later than the initial notes. However, our framework can be implemented such that Supernote candidates are regenerated when new information is available and a Supernote is posted as soon as the best candidate exceeds a certain helpfulness threshold. Finally, many misleading posts include images and videos which our current framework does not consider when generating and scoring Supernotes. Despite this limitation, the Supernotes were rated as significantly more helpful than the best existing community notes in our human subjects experiment.

**Future Work.** One exciting direction for future research is to expand our framework to incorporate external sources when generating Supernotes. For instance, the content of the web pages linked in the existing notes could be utilized when generating and scoring Supernote candidates. The principle alignment component could also be enhanced to verify that the linked sources indeed support the expressed claims. Beyond the links in the existing notes, the framework could automatically search for other reliable sources that offer different perspectives and further context. The main challenge of such extensions is ensuring that these external sources do not introduce any inaccuracies that could undermine users' trust in the system. Finally, it is important to carefully consider certain factors before deploying our framework. For example, Community Notes users earn "writing impact" when their notes are rated as helpful and displayed on the platform. An appropriate credit allocation system must be designed to fairly distribute the writing impact to users who contributed the notes included in the Supernote. This credit allocation can serve as a reminder that the system is intended to enhance, rather than replace, human contributions.

## 7 ETHICAL USE OF DATA

In this study, we conducted human-subjects experiments to evaluate the helpfulness of the Supernotes generated by our framework. Our protocol was reviewed by our institution's IRB and received an "exempt" determination. To train our personalized helpfulness models, we used publicly-available Community Notes data. This data is anonymized by design and cannot be linked to corresponding X users who contributed the notes and ratings.

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

# A PERSONALIZED HELPFULNESS MODEL (PHM)

## A.1 Sampling technique in the PHM

We train the Personalized Helpfulness Model (PHM) as a 10-layer neural network on publicly available data from Community Notes. At the last layer of the neural network, we use probabilistic sampling (as opposed to the more commonly used greedy sampling). Greedy sampling results in an over-representation of "helpful" ratings removing the uncertainty expressed through raw model output probabilities. When aggregated using the Community Notes algorithm, a vector of all "helpful" ratings correspond to a score of around 0.6 (Figure 9). Probabilistic sampling solves this issue and also results in a lower absolute error when compared with observed note intercept scores from Community Notes (Figure 9).

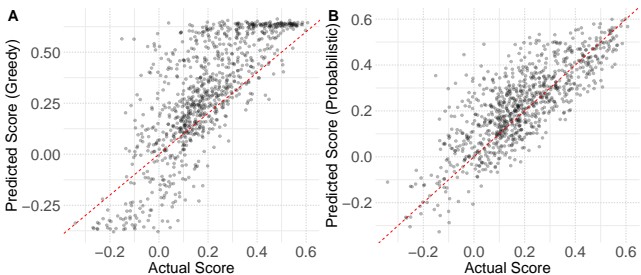

Figure 9: Impact of sampling strategies on the PHM. (A) Using greedy (argmax) sampling, the PHM predictions and note intercept scores are positively correlated but may sometimes inflate the aggregated score, resulting in an artificial ceiling at $y = 0.6$ and a higher error ($MAE = 0.16$). (B) Using probabilistic sampling (weighed by output probabilities) resolves the inflation of scores and results in a stronger positive correlation with observed note intercepts, resulting in lower absolute error ($MAE = 0.09$). The red dashed line represents perfect predictions.

## A.2 Output probabilities of the PHM

Probabilistic sampling (as described in A.1) relies more strongly on the actual values of raw logits from the model output, than greedy sampling, which selects the output label corresponding to the highest probability. To ensure that the probabilities predicted by the PHM are well calibrated, we plot a reliability curve [28] (Figure 10) and confirm that the probabilities aren't skewed for either of the 3 classes.

# B RESEARCH METHODS

## B.1 Matrix Factorization

We use the PHM to predict the scores the candidate Supernote would receive from each of the jury members. These scores are then aggregated to get a single score for each candidate. While a simple approach to aggregation could use the mean of these scores [15], we choose to adopt the aggregation function used in the Community Notes algorithm. This not only reduces false positives (as aggregated scores using this approach are typically lower than mean scores)

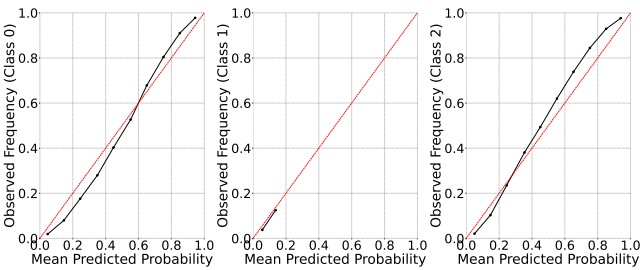

Figure 10: Reliability Curve of the PHM. Red line represents perfect calibration. We observe that the reliability curve is close to the diagonal implying that there is no systematic under-confidence or over-confidence in the PHM.

but also allows us to use existing thresholds (i.e., scores above 0.4 are considered helpful) to interpret expected helpfulness of these notes for Community Notes users. Below, we describe exactly how we compute this aggregation score.

User ratings in Community Notes are modeled as a combination of a global baseline (global intercept $\mu$) the user's unique rating tendency (user factor $f_u$ and intercept $i_u$), and the note's quality (note factor $f_n$ and intercept $i_n$). The primary component that explains much of the variation in ratings is the dot-product between the user and the note factors ($f_u \cdot f_n$), which captures how a user perceives a note's quality. The note intercept is interpreted as the note's helpfulness, representing the residual quality after accounting for the user's rating tendencies and the global baseline [42].

Using this model, we aggregate the PHM-predicted ratings of each note by solving for the note factor and intercept that best explain the ratings, considering the user features ($f_u, i_u$) of all jury members who rated the note. Since we typically have more ratings than unknowns (i.e., the note characteristics), we use a least-squares approximation to find the note intercept that best fits the PHM ratings. Finally, the note with the highest intercept is selected as the Supernote.

## B.2 Tag Questionnaire

As a part of the survey described in Section 4.1, users are asked to rate their agreement on five statements that correspond to the tags raters may select on Community Notes. These descriptive rating tags may be grouped into five broad categories: Source Quality, Clarity, Relevance, Veracity and Bias in Language. Below, we list the five statements as they appear in the survey along with the tags from Community Notes they represent:

(1) **Source Quality**:
   **S1:** The sources on the note are high-quality and relevant
   (a) Cites high quality sources
   (b) Sources not included or unreliable
   (c) Sources do not support note
(2) **Clarity**:
   **S2:** The note is written in clear, correct language
   (a) Easy to understand
   (b) Typos or unclear language
(3) **Relevance**:
   **S3:** The note addresses all key claims in the post

    (a) Directly addresses the post's claim
    (b) Misses key points or irrelevant
  (4) **Veracity**:
    **S4:** The note provides important context
    (a) Provides important context
    (b) Note not needed on this post
    (c) Incorrect information
  (5) **Bias in Language**:
    **S5:** The note is NOT argumentative, speculative or biased
    (a) Neutral or unbiased language
    (b) Argumentative or biased language
    (c) Opinion or speculation

## B.3 LLM Prompts

In our implementation of the framework, we leverage two templates for prompting an LLM. These are (a) to summarise existing notes into a single Candidate Supernote in a way that follows principles of effective fact-checking, and (b) to check whether a given Candidate Supernote adheres to a single principle of good fact-checking. The prompts as used are presented verbatim below:

(a) ```
X has a crowd-sourced fact-checking program, called
Community Notes. Here, users can write 'notes' on
potentially misleading content. Each note needs
to be rated by enough number of diversely-opinionated
people (note-raters) for it to be shown publicly
alongside the piece of content.
Your job is to craft a 'supernote' summarising
main points from existing notes (which I will
provide). This supernote should be able to replace
all existing notes. The goal of the supernote is
to maximise consensus among diversely opinionated
note-raters. It should be in unbiased language,
not argumentative, cite high-quality sources (links)
whenever applicable and should not add/ make-up
new facts. It should also be within 280 characters.
Post: <post text>
Note 1: <note text> (Tags: <tag 1>,<tag 2>, ...)
Note 2: ...
```

(b) ```
Answer with a 1(Yes) or 0 (No). Is this text
<fact-checking principle>?
```

All prompts were run on gpt-4o-mini-2024-07-18 with default parameters unless specified otherwise.

