# OpenReview forum: "Supernotes: Driving Consensus in Crowd-Sourced Fact-Checking"
_ACM.org/TheWebConf/2025/Conference — WWW 2025 Poster_

### Official Review · Reviewer_x5k4 · 2024-11-12

**Novelty:** 4
**Technical Quality:** 4

**Review:**

Brief Overview

This paper proposes a framework for creating Supernotes, which uses AI-generated consensus comments to foster agreement among diverse users in crowd-sourced fact-checking systems. Supernotes synthesize multiple existing community comments, generates candidate comments using large language models (LLMs), and filter them through a scoring model trained on extensive historical community ratings to identify the comments most likely rated as helpful by users. Experimental results show that Supernotes are perceived as more helpful than the best existing community comments and outperform LLM-generated summaries across five dimensions: quality, clarity, coverage, context, and argumentation.

Advantages

1. Supernotes leverages LLMs to generate diverse candidate comments and a robust scoring model to filter comments that are likely to be rated as helpful by a wide range of users, addressing the issue of consensus difficulty in existing systems.
2. The framework includes two components—candidate generation and candidate scoring. A "virtual jury" simulates user ratings to assess the helpfulness of candidate comments and selects those that meet the key principles of effective fact-checking.
3. In user experiments, Supernotes were found superior to existing community comments in covering essential context, providing high-quality information sources, and using clear language while being less prone to external information interference.
4. In the virtual jury, a Personalized Helpfulness Model (PHM) predicts ratings based on note content and rater characteristics. It uses probabilistic sampling rather than greedy sampling sampling to maintain relative uncertainty, resulting in more unbiased and accurate Supernotes evaluations.

Disadvantages

1. The framework relies on existing community comments and ratings, meaning Supernotes may have delayed response times and may not address real-time misinformation as quickly.
2. Despite the experimental design's attempt to simulate platform usage, the study was conducted through external surveys, which may not fully replicate the actual user experience.

**Questions:**

Questions
1. Can the Supernotes generation framework effectively adapt to multimodal content, such as generating helpful comments on posts containing images, videos, or audio to foster user consensus?
2. Given that this paper focuses on Platform X, how does Supernotes perform across different social media platforms? Would specific adjustments or optimizations be necessary?
3. Has there been consideration of incorporating external reliable sources (e.g., Wikipedia, reference books) during annotation generation to further enhance Supernotes’ context and comprehensiveness?
4. Can the framework extend to fact-checking fields beyond political information, such as health or entertainment information, or adapt to languages other than English?
5. As the Supernotes framework relies on LLMs, could future enhancements involve integrating more natural language processing techniques or deep learning algorithms to improve performance? For instance, could sentiment analysis or relationship extraction modules be introduced to enrich background understanding? For posts aimed at inciting user emotions, could the framework generate appropriately moderated content based on user sentiment?

**Reviewer Confidence:**

4: The reviewer is certain that the evaluation is correct and very familiar with the relevant literature

**Scope:**

3: The work is somewhat relevant to the Web and to the track, and is of narrow interest to a sub-community

---

### Official Review · Reviewer_SN7o · 2024-11-24

**Novelty:** 6
**Technical Quality:** 6

**Review:**

In this work the authors developed an LLM-driven framework to synthesize existing community notes and generate new Supernotes from existing proposed notes. They evaluated the LLM-generated candidates with a novel scoring model, and ran a human subjects experiment to compare the LLM notes with best existing community notes for 100 sample posts. They also perform a followup ablation experiment comparing LLM generated notes against the summaries that come through the framework to see how much the framework contributes to the success of the notes.

This is an interesting work that contributes at least somewhat to the community, and I think with minor clarifying changes should be accepted.

**Strengths**:
* It’s a compelling and intuitive idea to synthesize the information described across multiple notes into one supernote
* The performance and validity of the system is evaluated appropriately in the survey (mimicking to some degree how it would be deployed)
* The paper is well-written and methodologically pretty clear.

**Weaknesses**:
* There’s a lot of focus on the PHM but not the principle alignment module, it is unclear what that adds. Similarly, the prompt in the appendix for the principle alignment module seems underdeveloped
* Not enough description of the survey and participants, which is minor but hurts reproducibility


**Specific comments**

* 145 - are all successful notes (that are attached) neutral in tone?
* 324 - Your prompt as described on the github & appendix does not seem to specifically supply a definition of the principle.
* 401 - Is there a reason you chose this neural network architecture?
* 542 - is this in regards to the posts in the final section of the survey or the 12 posts. Did each participant see the same 10 posts at the end?
* 777 - I’d be curious how much the principle alignment part played a role, given that it filtered out “relatively” few candidates

* For historical purposes it would be useful to note that the Community Notes program was formerly known as Birdwatch

* It would have been interesting to evaluate a supernote against an already converged helpful note

* How strict were the principle alignment rule-based criteria? Could some notes feasibly slip through?

* Did participants follow links?

* Including a formal description of the linear mixed-effects model in the appendix would help readers and additionally strengthen the reproducibility of your results.

* Could use more information about the survey & participants -- were they X users at the time? How many users were assigned to each post? Were they trained at all beforehand?

**Questions:**

Given the cutoff of a month before the experiment and the constraints to US politics -- when was the experiment run?

Did you try other templating strategies like in-context learning and chain of thought?

How robust do you think your system would be if X implemented changes to the underlying Community Notes system?

**Reviewer Confidence:**

3: The reviewer is confident but not certain that the evaluation is correct

**Scope:**

4: The work is relevant to the Web and to the track, and is of broad interest to the community

---

### Official Review · Reviewer_hi2W · 2024-12-02

**Novelty:** 6
**Technical Quality:** 4

**Review:**

The paper proposes Supernotes, a framework that summarizes community notes in X to provide better consensus. It generates candidates  and scores them using LLMs (and model trained from LLM embedding). The evaluation consists of a human evaluation on 12 X posts that contains best existing notes and Supernotes's notes.

Strength:
1. The idea and motivation are clear. The whole paper is easy to following.
2. The framework does not seem to have any flaws to me.
3. The evaluation results seem promising.

Weakness:
1. The evaluation does not explains what are the X posts and the contexts. Notice one can easily cherry-pick some challenging posts from X and the best existing notes would clearly be sub-par. Some explanations on the posts that are used in the experiments are needed.
2. Given it is a post summarization technique based on LLM. I'd like to compare its performance against plain LLM prompting. For example, I'd like to see some results from just putting all the X's notes on a post to GPT4-o-mini and ask it to summarize and see the results.

**Questions:**

Same as weakness.

**Reviewer Confidence:**

3: The reviewer is confident but not certain that the evaluation is correct

**Scope:**

4: The work is relevant to the Web and to the track, and is of broad interest to the community

---

### Official Review · Reviewer_k34w · 2024-12-02

**Novelty:** 4
**Technical Quality:** 6

**Review:**

This paper targets a new mechanism in X/Twitter for crowd-sourced fact-checking: following original posts, users can leave new notes (community notes). If the notes are rated helpful by many diverse users (beyond a threshold), the note(s) are attached to the original post.

However, many of the posts that have at least one community note did not have enough user ratings such that its community note is finally attached. The authors identified two possible reasons for that. First, the polarized ratings. Some users support the note, while others don't, leading to the unmet helpful threshold for that note. Second, a single note may not be able to fully present the *"essential context"* for the post.

Then the authors proposed the **Supernotes** framework. This framework tries to understand all of a post's community notes (via LLMs) and generate new notes. The new notes are evaluated by a candidate scoring model trained on millions of history ratings. Human participants were asked to evaluate whether the notes generated by the Supernotes were more helpful than the best existing note or than the LLM summary generated one. The results showed that the scoring model has relatively high accuracy (AUC=0.85 and P@1=0.7) for predictions and the human participants rated the notes generated by Supernotes as more helpful.

Strengths:

1. The studied problem is important. The LLM-generated Supernotes are especially helpful for summarizing the existing notes and are worth exploring further.
1. There are many interesting insights into the Community Notes system and the results from the human evaluation.

Concerns:

1. The generated notes are based on the information in the post text and random subsets of existing community notes. This method may introduce redundancy in computation and information, which is ineffective and inefficient (100 Supernote candidates are required for Candidate Generation).
1. In the case of the 280-character limit of X, it may not fully represent the information of the community notes (the second challenge described in the paper).
1. Back to the core idea of the Supernote, which is used to promote consensus among users. Since Supernote is based on the existing basis of community notes, it cannot be voted as helpful by other users before it is generated. I mean, Supernote may struggle to obtain a sufficient number of notes from the users to make it reach the helpfulness threshold -- (1) the attention paid to the post by the public can rapidly decrease during the post's life cycle; (2) many posts (that have at least one note) do not have enough traffic for having a note to be attached. In addition, combining fragmented notes may increase polarization.
1. Following the above concern, it may not be ideal to foster consensus among users for a controversial/polarized post. The core of such posts is the controversial parts, but both sides supported Supernotes may only represent unimportant and ordinary information. It is because the algorithms are consensus-oriented and -trained, and are aligned with first-order principles. They may select very boring notes rather than informative/intriguing but slightly controversial notes to let the Supernote be "helpful."
1. The meaning of the "diverse set of users" should be more specific.

**Questions:**

Please see the concerns.

**Reviewer Confidence:**

3: The reviewer is confident but not certain that the evaluation is correct

**Scope:**

3: The work is somewhat relevant to the Web and to the track, and is of narrow interest to a sub-community